# Advances in Unmanned Aerial System Remote Sensing for Precision Viticulture

**DOI:** 10.3390/s21030956

**Published:** 2021-02-01

**Authors:** Alberto Sassu, Filippo Gambella, Luca Ghiani, Luca Mercenaro, Maria Caria, Antonio Luigi Pazzona

**Affiliations:** Department of Agriculture, University of Sassari, Viale Italia 39, 07100 Sassari, Italy; asassu@uniss.it (A.S.); lghiani@uniss.it (L.G.); mercenar@uniss.it (L.M.); mariac@uniss.it (M.C.); pazzona@uniss.it (A.L.P.)

**Keywords:** UAS, vegetation index, 3D vineyard characterization, canopy height model, precision farming, precision viticulture, remote sensing, sustainability of resources, vineyard detection and segmentation

## Abstract

New technologies for management, monitoring, and control of spatio-temporal crop variability in precision viticulture scenarios are numerous. Remote sensing relies on sensors able to provide useful data for the improvement of management efficiency and the optimization of inputs. unmanned aerial systems (UASs) are the newest and most versatile tools, characterized by high precision and accuracy, flexibility, and low operating costs. The work aims at providing a complete overview of the application of UASs in precision viticulture, focusing on the different application purposes, the applied equipment, the potential of technologies combined with UASs for identifying vineyards’ variability. The review discusses the potential of UASs in viticulture by distinguishing five areas of application: rows segmentation and crop features detection techniques; vineyard variability monitoring; estimation of row area and volume; disease detection; vigor and prescription maps creation. Technological innovation and low purchase costs make UASs the core tools for decision support in the customary use by winegrowers. The ability of the systems to respond to the current demands for the acquisition of digital technologies in agricultural fields makes UASs a candidate to play an increasingly important role in future scenarios of viticulture application.

## 1. Introduction

Precision agriculture concerns the use of multiple technologies to manage the spatial and temporal variability associated with agricultural production, improving crop performance, economic benefits, and environmental quality by limiting the use of pollutants [1,2,3]. In viticulture, precision agriculture techniques are used to improve the efficient use of inputs (e.g., fertilizers and chemicals), yield forecasting, selective harvesting of grape quality, and agree with the real needs (e.g., nutrients and water) of each plot within the vineyard [4]. New technologies have been developed for vineyard management, monitoring, and control of vine growth. Remote and proximal sensors become reliable instruments to disentangle vineyard overall status, essential to describe vineyards’ spatial variability at high resolution and give recommendations to improve management efficiency [5].

In the last decades, the development of aircraft and satellite platform technologies for remote sensing increased the spatial resolution, temporal availability, and capability to describe plants’ biophysical features [6,7]. Aircraft remote sensing campaigns can be planned with greater flexibility, but they are difficult and expensive [8]. Satellite image acquisition of large areas saves a considerable time, but has a low and inadequate resolution for precision viticulture (PV) [9]. Possible cloud cover combined with fixed acquisition times (referring to the time needed for the satellite to complete its orbit and return to the field area) could limit the monitoring process, and not allow early detection during specific phenological phases of the crop. Di Gennaro et al. [10] demonstrate the effectiveness of the spatial resolution provided by satellite imagery, Sentinel-2, on a trellis-shaped viticulture, as demonstrated for other permanent crops. However, due to the discontinuous nature of vine rows, their moderate coverage, soil influences between rows, background and shade, vineyards pose a challenge for remote sensing analysis: remote sensing images should be processed to separate the pixels of the canopy from the background [11].

Among all the remote sensing technologies for spatial and temporal heterogeneity detection, unmanned aerial systems (UASs) are the newest tools and likely the most useful in terms of high accuracy, flexibility, and low operational costs [12]. UASs can cover large rural areas much faster than people scouting on the ground, making it easier and more efficient to detect problems. UASs are often combined with imaging sensors, which allow the acquisition of images at higher spatial resolutions than those offered by satellites. Post-processing techniques combined with machine learning tools evolved to the point that the visual indications contained in an image can be extracted and transformed into useful information for farm management [13]. Poor weather conditions reduce the radiometric quality of the images resulting in less accurate and precise surface reconstruction. Reduced light conditions influence the stability of images’ features and increase errors in photo alignment and point cloud creation [14]. Calibration targets and post-processing techniques help standardize photo light conditions, especially in cloudy sky, low light conditions [15]. UAS remote sensing is a useful option for crop mapping even under cloudy conditions when satellite or airborne remote sensing are inoperable. The remote sensing task currently accounts for the majority of the operations performed with agricultural UASs [16]. In addition to applications involving the use of sensors and the extrapolation of useful information, UASs are applied and are under study for various types of operations, such as crop spraying operations [17,18,19,20,21,22], or combined with wireless sensor network (WSN) ground monitoring systems [23].

Compared to manual analysis processes characterized by high costs and operating times, the photogrammetry technique is a promising approach in precision farming scenarios, capable of creating more realistic models of crop structure, useful in decision-making processes. Andújar et al. [24] described an economic comparison of operating costs of different technologies for crop volume estimation. The results showed similarity in volume values, although on-ground technology provided a greater level of detail. However, the on-ground data acquisition costs were higher than that of aerial imagery. The obtained maps were used to perform a site-specific fertilizer spraying application, which involves a drastic reduction (80%) of product. 

Viticulture has shown the greatest technological advances among all agricultural sectors thanks to the higher profit margin resulting from the production of high-quality wine. In this scenario, UASs have proven to be profitable, reducing inputs and the environmental impact of agricultural activities.

Of the 395,000 European UASs expected to be deployed by 2035, SESAR [25] estimated that 150,000 will be used exclusively in the agricultural sector. The use of UASs in agricultural scenarios is well established [26,27], but currently there is no state of the art focused solely on the specific UAS remote sensing application in viticulture. The present work aims at providing a comprehensive overview of the application of UASs in PV, focusing on analysis of the sensors used, data extraction, analysis methods, and discusses the potential of UAS remote sensing as a management tool in viticulture scenarios. 

## 2. Unmanned Aerial Systems (UASs) Application in Viticultural Scenarios

The review is organized according to the research objectives stated above. The following sections explore the adoption of UAS remote sensing technology for different purposes in viticulture. 

### 2.1. Rows Segmentation and Crop Features Detection Techniques

The detection of intra-vineyard variability for site-specific interventions has always been a priority for PV, allowing grape growers to manage vineyards more efficiently and pursue a better grape production and quality. Satellite technology is not always able to guarantee a proper resolution to detect and differentiate the vine rows’ vegetation contours due to its coarse ground resolution. UASs, rather, show a high potential thanks to the sensor’s high resolution, with a ground sampling distance (GSD) often close to 1 cm. Vegetation indices are taken as useful tools for vegetation characterization, usually obtained by arithmetic spectral band combination [28]. Spectral information usually derives by visible red–green–blue (RGB), multispectral, hyperspectral, and thermal sensors mounted on board UASs [29,30,31]. Many vegetation indices have been used and compared for canopy biophysical estimation, including leaf area index (LAI), productivity, and biomass [32,33]. Matese et al. [34] proved the effectiveness of developed open source/low-cost UAS in real field conditions for vigor areas mapping within vineyards. The RGB images are useful tools for vineyard spatial variability monitoring, which requires an accurate segmentation to extract relevant information. Manual segmentation (e.g., by geographic information system—GIS) of RGB images is laborious, time-consuming, and needs to be improved to consider accuracies of the canopy, the shadow effect, and different soil conditions in inter-rows. Starting from ultra-high-resolution RGB imagery obtained from UAS, C. Poblete-Echeverría et al. [35] presented a vine canopy detection and segmentation approach using four different classifications methods (K-means, artificial neural networks, random forest, and spectral indices). The results showed how the 2G_RBi spectral index (derived by the difference in the divergence of the red and blue bands from the green in the absolute brightness of the channel), complemented with the Otsu method for thresholding [36], was the best option in terms of performance for vine canopy detection. This method was automatic and easy to apply since it does not need specific software to perform the calculations of the indices.

The high-resolution UAS images represent a challenge for classification due to higher intra-class spectral variability. In this spectral variability, object-based image analysis (OBIA) emerged in remote sensing segmentation applications [37,38]. The research carried out by Jimenez-Brenes et al. [39] aimed to develop a rapid mapping technique and obtain management maps to fight against the *Cynodon dactylon* (a typical vineyard weed). Starting from RGB and red–green-near-infrared (RGNIR) images, the team worked on the optimum spectral vegetation index, which is useful to classify bermudagrass, grapevine, and bare soil areas through an automatic algorithm, and the design of site-specific management maps for weed control. The geometric characteristics of the canopy are used in agriculture as a proxy of pruning, pest effects on crops, or fruit detection [40], but the collection of these data at the field scale is time-consuming and offers uncertain results. Despite the great variety of technologies used to characterize the 3D structures of plants (radar, digital photogrammetric techniques, stereo images, ultrasonic sensors, and light detection and ranging sensors), many of them have aspects that limit their use. Most of them are expensive, and it is challenging to use them in large spatial extents. The novelty of the work from Mesas-Carrascosa et al. [41] lies in the possibility to apply vegetation indices to RGB point clouds for the automatic detection and classification of vegetation and to determine grapevines’ height using the soil points as a reference. This automatic process, without any selected parameter of training, guarantees the lack of errors due to manual intervention in the separation process of the points’ classes.

As mentioned before, the extraction of pure vines pixels (i.e., the pixels that compose the leaf wall of the vines) is indispensable to achieve effective and good quality vineyard maps for site-specific management [42,43]. Comba et al. [44] designed a new methodology, constituted by three main steps based on dynamic segmentation, to identify vine rows from UAS aerial images even in the presence of low illumination, inter-row grassing, trees shadows, or other disturbance elements. The process works without any user intervention, and with a limited number of parameters for the calibration. The information obtained from this approach can be used in PV scenarios to obtain vigor and prescription maps for crop management or inter-row route tracking for unmanned ground vehicles (UGVs). Nolan et al. [45] described an automated algorithm, applied to a high-resolution aerial orthomosaic, for an unsupervised detection and delineation of vine rows. The algorithm takes advantage of “skeletonization” techniques, based on an extraction of a simplified shape (skeleton) of an object, to reduce the complexity of agricultural scenes into a collection of skeletal descriptors. Thanks to a series of geometric and spatial constraints applied to each skeleton, the algorithm accurately identifies and segments each vine row.

Pádua et al. [46] showed a method to automatically estimate and extract Portuguese vineyards’ canopies, combining vegetation indices and digital elevation models (DEM) derived from UAS high-resolution images, to differentiate between vines’ canopies and inter-row vegetation cover. It proved to be an effective method when applied with consumer-grade sensors carried by UASs. Moreover, it also proved to be a fast and efficient way to extract vineyard information, enabling vineyard plots mapping for PV management tasks. In the paper from Cinat et al. [47], three algorithms based on HSV (hue, saturation, value), DEM, and K-means were applied to RGB and RGNIR UAS imagery, to perform unsupervised canopy segmentation without human support over three scenarios derived from two vineyards. The first P18 scenario corresponds to the survey operations conducted in 2018 on 1 ha of commercial *Barbera* cv. vineyard. The M17 and M18 scenarios refer to flights performed in 2017 and 2018 on a 1.4 ha *Sangiovese* cv. vineyard. The two vineyards differ for different rows and slopes orientation and different intra-row and inter-row spacing. The research team tested the ability of the algorithms to identify grapevines without human supervision introducing estimation indexes. The estimation indices were useful to define the algorithm’s ability to over or under-estimate vine canopies. The three algorithms showed a different ability to estimate vines but, in general, HSV-based and DEM algorithms were comparable in terms of computation time. The K-means algorithm, however, increased computational demand as the quality of the DEM increased.

While rows identification from UAS images saw relevant development in the last years, a missing plant method was not developed until the study by Primicerio et al. [48] with a new methodology for vine segmentation in virtual shapes, each representing a real plant. They discussed, extracted, and coupled to a statistical classifier, an extensive set of features to evaluate its performance in missing plant detection within the parcels. Baofeng et al. [49] discovered instead the possibility to obtain accurate information about the affected or missing grapevines from a digital surface model (DSM). The analysis process started with a three-dimensional (3D) reconstruction from the RGB images, collected using the UAS, and the structure from motion (SfM) technique to obtain the DSM. A different approach followed by Pichon et al. [50], which did not involve the use of computer image analysis techniques, aimed at identifying relevant information that growers and advisers can extract from UAS images of the vineyard. The proposed methodology demonstrated that most of the information on grapevines status could be extracted from UAS-based visible images by the experts, assuming this information of great interest throughout the growing cycle of the vine, particularly for advisers, as support to drive management strategies.

### 2.2. Vineyard Remote Analysis for Variability Monitoring

PV could be defined as the set of monitoring and managing for spatial variability in physical, chemical, biological variables related to the productivity of vineyards [51]. A primary work, about the UAS platform and implemented sensors for data collecting, was carried out by Turner et al. [52] showing the perspective of the UAS technology to provide “on-demand” data. They analyzed the algorithms used in data processing, in the orthorectification process, and the vegetation indices to evaluate the differences within the vineyard images. The results highlighted the potential of UAS multi-sensor systems in PV, and their versatility enhanced by the possibility to collect data sets “on-demand” with a temporal resolution that spans the critical times in the crop growing season. The UASs spatial resolution permits to collect imagery at a much higher resolution and investigate a bigger spatial variability inside the vineyard compared to satellites and aircraft [53]. Differently from satellite technology, limited due to unfavorable re-visit times and orbit coverage patterns [54], UAS close-range photogrammetry represents an efficient method for continuously collecting information [55]

Matese et al. [56] introduced a new technique to evaluate the spatial distribution of vine vigor and phenolic maturity. A normalized difference vegetation index (NDVI) map was obtained by a high-resolution multispectral camera mounted on a UAS. Spatial variability of grape anthocyanin content was detected in situ evaluating ANTH_R and ANTH_RG indices by using a fluorescence-based sensor (Multiplex^TM^). The two techniques appeared suitable to compare vine related information on a relatively large scale. The research by Zarco-Tejada et al. [57] showed the feasibility of mapping leaf carotenoid concentration from high-resolution hyperspectral imagery. The R515/R570 index was explored for vineyards in this study. The PROSPECT-5 leaf radiative transfer model was linked to the SAILH and FLIGHT canopy-level radiative transfer models to simulate the pure vine reflectance without soil and shadow effects due to the UAS hyperspectral imagery, which enabled targeting pure vines. Primicerio et al. [58] used a UAS as a tool to combine high spatial resolution images, quick turnaround times, and low operational costs for vegetation monitoring, providing low-cost approaches to meet the critical requirements of spatial, spectral, and temporal resolutions needed. A low cost and open-source agro-meteorological monitoring system was designed and developed, and its placement and topology were optimized using a set of UAS-taken multispectral images. Mathews [59] captured aerial images of a Texas vineyard at post-flowering, veraison, and harvest stages using digital cameras mounted on board a UAS. The images were processed to generate reflectance orthophotos and then segmented to extract canopy area and NDVI-based canopy density. Derived canopy area and density values were compared to the number of clusters, cluster size, and yield to explore correlations. Differently from the derived canopy area, the NDVI-based canopy density exhibited no significant relationships because of the radiometric inaccuracy of the sensors. A vine performance index (VPI) was calculated to map spatial variation in canopy vigor for the entire growing season. C. Rey-Caraméset al. [60] used multispectral and spectral indices to assess vegetative, productive, and berry composition spatial variability (obtained by SFR_R_AD_ and NBI_G_AD_ Multiplex^TM^ indices) within a vineyard. The correlations were significant but moderate among the spectral indices and the field variables, the pattern of the spectral indices agreed with that of the vegetative variables and mean cluster weight. The results proved the utility of the multi-spectral imagery acquired from a UAS to delineate homogeneous zones within the vineyard, allowing the grape-grower to carry out a specific management of each subarea. The aim of the work by Matese et al. [61] was to evaluate different sources of images and processing methodologies to describe spatial variability of spectral-based and canopy-based vegetation indices within a vineyard, and their relationship with productive and qualitative vine parameters. Comparison between image-derived indices from Sentinel 2 NDVI, unfiltered and filtered UAS NDVI, and agronomic features were performed. UAS images allow calculating new non-spectral indices based on canopy architecture that provide additional and useful information to the growers with regards to within-vineyard management zone delineation. Caruso et al. [62] identified three sites of different vines vigor in a mature vineyard to test the potential of the visible-near infrared (VIS-NIR) spectral information acquired from an UAS in estimating the LAI, leaf chlorophyll, pruning weight, canopy height, and canopy volume of grapevines. They showed that the combined use of VIS-NIR cameras and UAS is a rapid and reliable technique to determine canopy structure and LAI of grapevine. Romboli et al. [63] focused on the impact of vine vigor on Sangiovese grapes and wines, applying a high-resolution remote sensing technique by a UAS platform to identify vigor at the single vine level. The test confirms the ability of UAS technology to assess the evaluation of vigor variability inside the vineyard and confirm the influence of vigor on the flavonoid compounds as a function of bunch position in the canopy. Matese and Di Gennaro [64] described the implementation of a multisensory UAS system capable of flying with three sensors simultaneously to perform different monitoring options. The vineyard variability was assessed in terms of characterization of the state of vines vigor using a multispectral camera, leaf temperature with a thermal camera, and an innovative approach of missing plants analysis with a high spatial resolution RGB camera. 

Pádua et al. [65] developed an analysis methodology useful to assist the decision-making processes in viticulture. They employed UASs to acquire RGB, multispectral, and thermal aerial imagery in a vineyard, enabling the multi-temporal characterization of the vineyard development throughout a season, thanks to the computation of the NDVI, crop surface models (CSM), and the crop water stress index (CWSI). Vigor maps were computed first considering the whole vineyard, second considering only automatically detected grapevine vegetation, and third considering grapevine vegetation by applying a normalization process before creating the vigor maps. Results showed that vigor maps considering only grapevine vegetation provided an accurate and better representation of the vineyard variability, gathering significant spatial associations through a multi-temporal analysis of vigor maps, and by comparing vigor maps with both height and water stress estimation. The objective of the work by Matese et al. [66] was to evaluate the performance of statistical methods to compare different maps of a vineyard, derived from UAS acquired imagery, and some from in situ ground characterization. The team proved how these methods, which consider data spatial structure to compare ground autocorrelated data and spectral and geometric information derived from UAS-acquired imagery, are highly appropriate, and would lead winegrowers to implement PV as a management tool. Pádua et al. [67] developed a multi-temporal vineyard plots analysis method at a grapevine scale using RGB, multispectral, and thermal infrared (TIR) sensors, enabling the estimation of the biophysical and geometrical parameters and missing grapevine plants detection. A high overall agreement was obtained concerning the number of grapevines present in each row and the individual grapevine identification. Moreover, the extracted individual grapevine parameters enabled the assessment of vineyard variability in each epoch and to monitor its multi-temporal evolution.

TIR sensors mounted onboard of UASs allow the obtainment of important information about soil and crop status derived by the analysis of temperature spatial variability and the crop water stress detection inside the field [68]. More precise monitoring of water status based on thermal data is required, since this kind of information can be implemented in water management irrigation strategies, especially when dealing with different genotypes in the same vineyard [69]. Thermal and multispectral imagery could assess and map the spatial variability of water status inside the vineyard. The goal of the article by Baluja et al. [70] was the water status variability assessment of a commercial vineyard using thermal and multispectral imagery derived from sensors mounted on a UAS and comparing them with leaf stomatal conductance and stem water potential. The authors stated that the relationship between thermal imagery and water status parameters could be considered as a short-term response. NDVI and TCARI/OSAVI indices were probably reflecting the result of cumulative water deficits in a long-term response. Bellvert et al. [71], used the CWSI to map the spatial variability in water deficits across a *Pinot noir* cv. vineyard using a thermal UAS sensor in different hours of the day. CWSI was correlated with leaf water potential (LWP) determined by canopy temperature measurement with infrared temperature sensors. The biggest correlation between CWSI and LWP at 12:30 hours of the day suggests that the latter was the more favorable time for obtaining thermal images due to the correlation with LWP values. The sensitivity analysis of pixel sizes confirmed that a 0.3 m pixel was required for precise CWSI mapping. Sepúlveda-Reyes et al. [72] showed that it is necessary to consider grapevine architecture and image thresholding approaches for proper use of aerial and terrestrial thermography techniques. An experimental study under different water stress conditions was run in a commercial *Carménère* cv. vineyard trained with vertical shoot position (VSP). In this study, thermal images were obtained from different canopy zones by using both aerial and ground-based thermography. The standard deviation technique (SDT), the energy balance technique (EBT), and the field reference temperature (FRT) technique were used as different thresholding approaches to each image. Results obtained showed that the EBT had the best performance, discriminating over 95% of the leaf material. Ground-based nadir images presented the best correlations with stomatal conductance and stem water potential in the case of canopy zone analysis. The best relationships between thermal indices and plant-based variables were registered during the period of maximum atmospheric demand (near veraison), with significant correlations for all methods. The aim of the work from Santestebana et al. [73] was to evaluate how high-resolution thermal imaging can provide instantaneous and seasonal variability data of water status within a vineyard. A spatial modeling approach was used to test the potential of the CWSI acquired in a single day to estimate patterns of variation in water status within-vineyard. CWSI correlated well with stem water potential and stomatal conductance showing a great potential to monitor instantaneous variations in water status within a vineyard. Thermal images information proved to be relevant at a seasonal scale. A single day measurement using the CWSI did not provide a good estimation of variations of plant water status but simulated other physiological processes occurring during ripening. The aim of the study by Poblete et al. [74] was to develop artificial neural network (ANN) models derived from multispectral images to predict the stem water potential spatial variability of a drip irrigated *Carménère* cv. vineyard in Talca (Chile), useful for the assessment of vine water status variability. The obtained coefficient of determination between ANN outputs and ground-truth measurements of stem water potential was between 0.56–0.87, with the best performance observed for the model that included the bands 550, 570, 670, 700, and 800 nm. Poblete et al. [75] used the midday stem water potential (SWP), and TIR imaging for CWSI to assess grapevines’ water stress. An automatic co-registration of thermal and multispectral images, obtained from a UAS, was employed to remove shadow canopy pixels by a modified scale-invariant feature transformation (SIFT) computer vision algorithm, and K-means++ clustering. The proposed methodology improved the analysis of the relationship between CWSI and SWP by shadow canopy pixels removing from a drip-irrigated vineyard. The study showed a higher negative effect of shadow canopy pixels in grapevines affected by water stress, compared with well-watered vines. Tucci et al. [76] performed a thermal characterization of a dry-stone wall terraced vineyard using a visible and thermal infra-red sensor, to detect possible microclimate influence derived by dry-stone terracing. The results revealed the different behavior of the rows during the morning and the afternoon.

### 2.3. Rows Area and Volume Estimation

Plant architecture and variation of the surface area and volume occupied by the foliage are useful characteristics for the characterization, monitoring, and protection of the vineyard production. The precise knowledge of area and volume occupied by the rows would help farmers to detect the vegetative development of the crop, identify deficiencies, and optimize canopy treatments by implementing a site-specific crop management system. This approach can optimize and limit the inputs and diminish potential environmental damages caused by an inappropriate application of products, as well as reduce the management cost. Direct methods for canopy structure analysis are extremely time-consuming and hardly applicable at a large-scale. The advances of using UAS for remote sensing and photogrammetry application opened a new way to obtain rapidly, and on a large scale, this information, and to combine 3D measurements with spectral information.

The study from Mathews and Jensen [77] is a preliminary study of how structure from motion algorithm (SfM) can help to predict quickly, practically, and inexpensively the LAI of a vineyard using extracted points from a point cloud. This work represented one of the first reasonable successes of this method showing the practical and inexpensive nature of the SfM method of 3D. Kalisperakisa et al. [78] compare and evaluate LAI estimation in vineyards from different UAS imaging datasets (hyperspectral data, 2D RGB orthomosaics, and 3D CSM. The overall evaluation indicated that the estimated canopy levels were correlated with the in-situ, ground truth LAI measurements. The highest correlation rates were established with the hyperspectral canopy greenness and the 3D CSM. The lowest correlations, instead, derived from the calculated greenness levels of the 2D RGB orthomosaics. The study by Pádua et al. [79] aimed to characterize vineyard vegetation through multi-temporal monitoring using a commercial low-cost UAS equipped with an RGB sensor. The used image-processing techniques enabled the extraction of different vineyard characteristics and the estimation of its area and canopy volume, providing a quick and transparent way to assist winegrowers in managing grapevine canopy. Ballesteros et al. (2015) [80] used images obtained from UAS to characterize the growth attributes of LAI, green canopy cover (GCC), and canopy volume (CV) of irrigated and rain-fed vineyards in semi-arid conditions. The results showed that the behavior of the foliage is determined by the characteristics of the cultivar, the training system, pruning practices, and cultural practices. The relationships between LAI and growing degree days (GDD), CV and GDD were used to determine a reliable canopy structure model during the growing season and contribute to the optimization of site-specific management within the vineyard. Biomass is an important variable, useful to choose suitable canopy management within the field, and it can be estimated using plant canopy height. Crop surface models can be used to obtain plant height, in combination with a non-vegetation ground model. The traditional biomass estimation method used by the farmer is often imprecise and time-consuming compared to non-destructive and fast crop surface models (CSMs) estimation by UASs’ imagery. L. Comba et al. [81] have tested the reliability of an estimation process of a dense 3D point cloud as an economical alternative to traditional LAI assessments. The LAI was estimated using a multivariate linear regression model that uses 3D crop crown descriptors (thickness, height, and distribution of leaf density along the wall), showing a high correlation with those obtained with the traditional manual method, even in hilly and difficult-to-access regions. Matese et al. [82] constructed a 3D DSM for the creation of precise digital terrain models (DTM), to be subtracted from the DSM to obtain a canopy height model (CHM) of the vineyard. The results showed good separation between ground pixels and vine rows, but their height was not quite following the actual height of the vines, due to a smoothing effect attributable to low camera resolution. A further comparison between CHM and a vigor map obtained from the NDVI values showed a good correlation. The average canopy height and vine row width were used for a preliminary assessment of biomass volume. The work by Pichon et al. [83] focused on the assessment of the quality of low-cost DSMs obtained with UAS images (provided by three companies), and test whether the DSMs met common requirements of the wine industry. The DSMs quality were analyzed through the mean error and its dispersion in the XY plane and in elevation Z. The results showed good quality DSMs, able to assess field characteristics of elevation, slope, and aspects, useful for terroir characterization. The study proved the efficiency and the reliability of elevation data derived from UASs, with an accuracy equivalent to the reference system used in the study. De Castro et al. [84] developed an object-oriented algorithm using photos obtained from a low-cost RGB camera mounted on board a UAS for 3D characterization of the vineyard. The OBIA algorithm was able to measure the volume and height of grapevine canopy and detect missing plants without previous training or human assistance, adapting itself to different field conditions. Ronchetti et al. [85] focused on the production of high-resolution DTM in agriculture by photogrammetric processing fisheye images, acquired with very light UAS. Different flight strategies have been tested together with different GCPs and check point (CP) configurations and software packages. The computed DTMs have been compared with a reference model, obtained through geostatistical analysis (using Kriging interpolation) of GNSS-RTK measurements. The photogrammetric DTMs showed a good agreement with the reference one. Ghiani et al. [86] described the development of a methodology for the computation of the canopy volumes through remotely sensed imagery acquired with UAS RGB digital camera, analysis with MATLAB scripts and ArcGIS. Preliminary results showed that the volumes obtained with this 3D reconstruction were 50% lower than those directly measured in the field by the tree row volume (TRV) technique, therefore promoting a limitation of the use of chemicals. The paper from Burgos et al. [87] presented the acquisition methodology of high-resolution images using UAS and their processing to construct a 3D DSM and a DTM. The subtraction of DTM from the DSM permitted to obtain a differential digital model (DDM) of a vineyard, in which the pixels with an elevation higher than 50 cm above ground level were detected as vine pixels. The results show that it was possible to separate pixels from the green cover and the vine rows, with DDM values between −0.1 and +1.5 m. Starting from RGB color model imagery obtained by UAS, M. Weiss and Baret [88] proposed a methodology to describe vineyard 3D macro-structure using the dense point cloud to distinguish the background from the vineyard and applying a threshold on the height to separate the rows from the row spacing. The quality of the dense point cloud seems to affect the row width, cover fraction, and the percentage of missing row segments. Comba et al. [89] proposed new and more reliable methods for vineyard monitoring operations, using a data fusion approach for vigor characterization in vineyards. It exploits the information provided by 2D multispectral aerial imagery, 3D point cloud crop models, and aerial thermal imagery of 30 portions of vine rows. The data fusion methodology showed how vigor could be automatically evaluated by a generic 3D point cloud, without any user intervention or manual vineyard boundaries selection. Comba et al. [90] also proposed an innovative unsupervised algorithm for vineyard detection and vine-rows features evaluation, based on 3D point-cloud maps processing The main results are the automatic detection of the grapevines (at different phenological phases and growth stages), the local evaluation of vine rows orientation, the inter-rows spacing in the presence of dense inter-row grassing, the detection of missing plants, and steep terrain slopes. The effectiveness of the developed algorithm did not rely on the presence of rectilinear vine rows, being also able to detect vineyards with curvilinear vine row layouts. One of the last works by Comba et al. [91] involved the generation of low complexity 3D mesh models of vine rows from 3D point clouds, reducing the number of georeferenced instances to describe the spatial layout and shape, with a reduced amount of data (98%), without losing relevant crop shape information. This process will facilitate the computational process and allows a real-time interpretation of point clouds. The 3D process for VSP-training systems can automatically process in hilly areas and non-uniform vineyards characterized by non-linear vine rows and different intra-row distance.

### 2.4. Crop Disease Detection

There are multiple grapevine diseases responsible for yield quality and quantity decrease and economic losses to the wine industry worldwide. Symptoms can be evident, and in some cases completely absent. For this reason, a continuous monitoring of the plants and the detection of symptoms becomes fundamental to preserve the health of the vineyard and protect the harvest. 

Di Gennaro et al. [92] suggests a methodology to investigate the relationships between high-resolution multispectral images (0.05 m/pixel) acquired using a UAS, and grapevine leaf stripe disease (GLSD) foliar symptoms monitored by ground surveys. This approach showed high correlation between NDVI index and GLSD symptoms, and discrimination between symptomatic and asymptomatic plants was surveyed and mapped since 2003. Each vine was located with remote sensing and ground observation data were analyzed to promptly identify the early stages of the disease, even before visual detection. This work suggests an innovative methodology for quantitative and qualitative analysis of the spatial distribution of symptomatic plants. Albetis et al. [93] presented the potential of spectral bands, vegetation index, and biophysical parameters to detect grapevines affected by the *Flavescence doree* disease caused by the bacterial agent *Candidatus Phytoplasma vitis* and transmitted by the insect *Scaphoideus titanus* (*Ball*). They used the receiver operator characteristic (ROC) analysis to determine the ability of each variable and found that RGI and GRVI vegetation indices based on the green and red spectral bands were the best to detect *Flavescence doree* pixels in UAS multispectral imagery. In another study, del-Campo-Sanchez et al. [94] quantified the impact of the pest *Jacobiasca lybica* on vineyards and developed a representative cartography of the severity of the infestation. To accomplish this work, computational vision algorithms based on an ANN combined with geometric techniques were applied to geomatic products using consumer-grade cameras in the visible spectra mounted on board a UAS. The results showed that the combination of geometric and computational vision techniques with geomatic products generated from conventional RGB images improved image segmentation of the affected vegetation, healthy vegetation, and ground. Thus, the proposed methodology is a more cost-effective application of UASs compared with multispectral cameras and increases the accuracy of estimations for the impact of pests by eliminating the soil effects. Vanegas et al. [95] implemented a UAS remote sensing-based methodology for the development of a predictive model for *Viteus vitifoliae* (*Fitch*, 1856) phylloxera detection. They explored the combination of airborne RGB, multispectral, and hyperspectral imagery with ground-based data at two separate time periods and under different levels of infestation. The investigated indexes could be used to determine the extent of the disease, the severity of the plant pest, and its impact on grape production. Besides, it could become a strategic decision support system (DSS) tool for vineyard management and could improve the potential for early detection of the pest.

### 2.5. Prescription Maps for Spraying Management

The plant-protection products application is a key aspect associated with environmental contamination, safety of operators, food safety, and the economical balance of crop production [96]. Every crop is distinguished by a structure, dimensions, and even foliar area and density. A crucial aspect directly related canopy characteristics regard the most efficient amount of pesticide, and the optimal amount of water to be applied. Canopy characterization becomes then a crucial aspect for what is defined site-specific management strategies. UASs potentially rely on the capability to characterize large areas, with relatively low cost, a great capability for recording large volumes of data, and potential to obtain a real picture from above, giving complementary information about crop distribution over the measured area.

Campos et al. [97] showed how canopy maps, obtained from a multispectral camera embedded in a UAS and variable rate application over a vineyard parcel, can potentially save wastes in pesticide application, water use, and time. Thanks to specific software, the prescription map was uploaded into a modified sprayer for the variable application process. Excellent accuracy was obtained with the system, saving water and pesticide by over 40%. Khaliq et al. [98] compared the decametric satellite resolution of Sentinel 2 and 35 m altitude UAS multispectral imagery of a vineyard using three different NDVI indices, considering the whole cropland surface, only the crop canopy pixels, and only the pixels representing the inter-row terrain. The pixels contained in the photo obtained from the UAS representing crop canopies inside the vigor maps resulted better related to the in-field assessment, compared to the satellite imagery. This approach showed how satellite imagery is unusable for crops grown in rows, for crop canopies which do not extend to the whole surface, or where the presence of weeds is significant. Campos et al. [99] designed a procedure for a variable rate application (VRA) sprayer. An unmanned aerial vehicle, equipped with a multispectral camera, was used to generate a photo set for the canopy characterization throughout the entire growing season in four vineyard plots. The multispectral images were then merged with the information provided by a DSS to obtain the prescription maps and apply the optimal volume rate. The prescription maps were then uploaded to the VRA prototype, obtaining updated maps after the application processes were complete. The prototype had an adequate spray distribution quality, with coverage values in the range of 20–40%, and exhibited similar results in terms of biological efficacy on *Plasmopara viticola* (*Berk. & M. A. Curtis*) *Berl. & De Toni* 1888 (powdery mildew) compared to conventional (and constant) application volumes.

## 3. UAS Platforms, Sensors, and Targets

The research studies analyzed in this review showed increased use of multi-rotor UASs instead of fixed-wing UASs. Fixed-wing and multirotor are the most common types of UASs. Both systems show the advantages and disadvantages of their use. The multirotor is easy to fly, take off, land, and operate thanks to a high number of dedicated software and applications. The major multirotor limitation concerns the reduced flight range, which leads to a reduction in the area that can be analyzed in a single flight. Fixed-wing UASs require a suitably large, obstacle-free landing area (often missing in vineyards) and good piloting skills to land them smoothly and avoid damage to the UAS and the sensors installed onboard. However, they differ in their excellent flight endurance and remarkable range that allows them to cover large areas in a single flight. Factors such as maintenance and flight time range favor the use of a fixed-wing, whereas the multi-rotor is preferred for proximity inspections and if more detailed and accurate data are required [100]. Thanks to the gimbal system, usually implemented onboard of the multirotor, the sensors can give a different 45° perspective (with respect to the ground surface) of the rows, so it is possible to obtain more reliable and useful data for 3D reconstruction of the rows, estimate their volume, or detect the presence of phytopathology on the leaf wall [50]. As shown in Table 1, most of the systems used were quadcopters, hexacopters, and octocopters. Their use is likely due to the limited size of the areas studied that do not involve long flights. Fixed-wing UASs appear in a limited number of papers. This choice stems from the need to cover large areas in a single flight. Vineyard surfaces reported in Table 1 range from a minimum of 0.3 ha to a maximum of 12 ha, with three exceptions of 14.0 ha, 17.7 ha, and 23.2 ha [44,87,98]. Fixed-wing UASs flew over the largest areas [45,71,87,88]. Differently from the satellite imagery, the investigation surface extension depends on the flight altitude, UASs speed, sensor’s field of view, and from the limited battery autonomy of multirotor UASs (on average 30 min) [101]. The vineyards showed heterogeneous characteristics in terms of slope, sun exposure, and distance between and within rows. 

All research groups used commercial UASs. The only customization made concerns the type of sensor on board. This choice is probably dictated by the high performance of commercial UASs, able to guarantee the necessary research functions at a lower purchase cost than custom UASs.

Most of the monitoring operations were carried out using RGB and multispectral sensors; only a small number of them involved thermal sensors and even less hyperspectral technologies. Because of the used sensors, most of the investigated spectral bands regarded the visible light (red, green, and blue) and the near infrared (NIR) band; only in a few cases the red edge (RE) and thermal infrared (TIR) bands were investigated. Hyperspectral sensors are not commonly used in remote sensing viticulture, their purchase cost is particularly high, and the data obtained are difficult to interpret. Different types of sensors capable of picking up different spectral bands were used during the field tests. As can be seen in Table 1, there are many manufacturers that currently produce sensors specifically designed to perform remote measurements via UAS. However, in several works, we can notice the use of reflex and mirrorless cameras, normally used in amateur and professional photo shoots. Increasing investments are expected in the development of parallel technologies and specific sensors for remote sensing in agriculture [102]. From this literature review, it emerged that most of the studies had the analysis of space–time variability within the vineyards as their main objective. As shown in Table 1, this type of approach is prevalent with respect to techniques for detecting canopy characteristics and calculating the structural characteristics of the canopy. The detection of plant diseases and the creation of vigor and prescription maps have been investigated in less depth. These results show the real interest of researchers in finding effective and fast methodologies to monitor vineyard development, understand the causes of this variability, and ensure precise, calibrated, and targeted crop management. The origin of intra-vineyard variability may derive from an inaccurate arrangement of the soil and water control plans, an imprecise setting of the grapevines during the training pruning phase, the non-application of some green and winter pruning techniques [103]. 

## 4. Perspective and Future Challenges

The vineyard ecosystem represents a major environment to investigate since grape and wine production is a proficuous economic activity. As pointed out in the introduction, the aim of PV regards the correct management of variability in the vineyard production system, the increase of economic benefits, and the reduction of environmental impact. The most relevant aspects concern the efficient use of inputs, the differentiation of grape qualities at harvest time, the prediction of yields, and greater accuracy and efficiency of sampling conducted at the plot level. Climate and meteorology are among the factors that influence vine productivity [104]. It is essential to understand how and how strongly climate and meteorology influence grape productivity and quality. Micrometeorology of vineyards can be implemented by examining thermodynamic variables, studying the exchange processes between soil, canopy, and atmosphere. Understanding how vineyard structure, composition, and farming practices can alter microclimate could help winemakers in their decision making process and management choices [105]. The availability of multiple data at the vineyard could allow the implementation of sophisticated algorithms to create phenological models, useful to predict the impact of climate change on the vine in the long term. The used tools allow monitoring of weather conditions above and within the vine canopy and in the soil. The conventional meteorological data and agrometeorological monitoring stations located far from the vineyard may not allow direct characterization of microclimatic or micrometeorological conditions, as the measurements are not representative of the actual physical conditions to which plants are subjected within vineyards [106]. Such conditions emerge when hourly data are investigated with the appropriate indices [107], whereas daily or monthly data tend to mitigate such differences. UAS technology could improve the acquirement of atmospheric data over extended surfaces in viticulture scenarios. Atmospheric information about pressure, temperature, humidity, and wind measurements can be easily and quickly obtained by UAS, as shown by [108]. The latest developments in automation could improve atmospheric monitoring operations with the required speed and frequency. When UAS data are not available or compromised by instrumental errors, ecosystem conditions should be provided by performing ground sampling operations or extracting proximal sensor data [109,110,111,112]. Further studies on this topic are needed to improve crop protection, highlight the damage to the ecosystem, and help farmers reduce the number of treatments.

More recent and relevant studies on the application of UAS remote sensing for soil moisture monitoring [113,114,115,116], water consumption and use efficiency [117,118,119,120,121], and surface energy budget [122,123] should be investigated to enrich future research directions in PV. These works demonstrate how the application of high-resolution remote sensing technology would improve knowledge of soil characteristics as a key element in site-specific vineyard management. 

From the bibliographic research on the use of UAS in viticulture, a limited amount of works focusses on the theme of in-flight product distribution operations. This shortcoming is justified by the innovativeness of these operations, as well as the evident limits to perform homogeneous distributions over the entire volume of the rows, and the limited autonomy of the UASs [21]. 

The high capacity of UASs to scan large areas and more fields during a crop season in a short period of time reduces operational cost. Due to still immature fully automated analysis procedures, the main cost associated with mapping is related to human labor for post-processing data elaboration. The use and diffusion of UAS technology depend unequivocally on the economic advantages derived by their application as decision support tools. UAS crop monitoring and mapping imply several logistic and business adjustments that result in site-specific management. More efforts are requested to evaluate the effective cost of UASs application, and to quantify the real economic benefits derived by the usage as DSS in viticultural fields.

## 5. Conclusions and Remarks

Our review provides a state of the art of UAS remote sensing in PV, focusing on the description of the applied methodologies and the obtained results. The cited studies differ for the different application purposes and the employed equipment, showing the potential of different technologies combined with UASs in the identification of variability of the vineyard through the characterization of structural characteristics, the presence of disease, and plant physiology. Therefore, UASs prove to be a technology that is well suited to different viticultural scenarios, sensors, and analysis techniques. As a result of the analysis of the cited works, there is a lack of studies related to the application of UAS remote sensing for plant disease detection and the creation of prescription maps for vineyard-specific treatment. The development of these topics is of great importance. They could lead to improvements in terms of the decision-making process. Improved disease detection and prevention, combined with prescription maps and variable rate application, will reduce product waste and more sustainable viticulture.

As a result of innovations in UAS technology, lower purchase costs, and an increasing use of such systems, UASs are a key tool for decision support in the customary use by winegrowers. This support can be enhanced thanks to a correct interpretation of data and their transformation into useful information to be integrated with proper agronomic management. Most of the work focused on the methodology of analysis and data acquisition. It would be useful to provide a series of works in which the described methodologies are applied in viticulture scenarios, comparing and partnering UASs with existing technologies to verify their actual effectiveness. Given the high level of training and expertise required, there is a clear need to develop simplified and more automated analysis methodologies for greater dissemination in real operational fields. The skills required for the interpretation of UAS images represent one of the key points for the development of the sector and go beyond flight planning and its implementation. The development of user-friendly software could be a turning point for the complete dissemination of these methodologies, still too difficult to be performed by those who do not have specific training.

The greater use of RGB and multispectral sensors compared to thermal, hyperspectral, and LIDAR chambers, shows a greater interest of research groups towards this type of sensor, probably due to a greater facility of data interpretation and a lower purchase price. The more accessible the technology, the more research teams can use it and the greater the growth of knowledge. The ability of the systems to respond to current demands for the acquisition of digital technologies in the agricultural field, candidates’ UASs, between all information and communication technologies, to play an increasing role in future scenarios of viticulture application. The multiple national regulations that govern and regulate the flight represent a system of safety and control of UAS operations. The increasing use of UASs in agricultural scenarios will benefit from regulatory simplification and unification among the various countries. UASs could be adopted in crop monitoring and management, improvement of crop productivity, optimization of crop resources, and reduction of operation time. They could be also employed through platforms for the management of swarms of UASs and put in communication with ground robots and tractors for specific operations.

## Figures and Tables

**Table 1 sensors-21-00956-t001:** Unmanned aerial system (UAS) typology used for viticulture research purposes and sensors information.

Spectral Range	Sensor Brand and Model	UAS Typology	UAS Brand/Model	Surface (ha)	Vineyard Cultivar	Objectives	Year	References
R-G-B-NIR-TIR	Canon 550D	Hexacopter	Mikrokopter He0xa-II	2	NA	variability monitoring	2011	[52]
R-G-B-NIR-TIR	MCA-6 Tetracam A40 M FLIR	Quadcopter	NA	5	Tempranillo	variability monitoring	2012	[70]
R-G-B-NIR-RE hyperspectral	Hyperspec VNIR	Fixed wings	mX-SIGHT	NA	Tempranillo	variability monitoring	2013	[57]
R-G-B	Canon PowerShot A480	Quadcopter	Hawkeye II	1.9	Tempranillo	rows geometry estimations	2013	[77]
R-G-B-NIR	ADC-lite camera Tetracam	Hexacopter	Mikrokopter Hexa-II	NA	NA	crop features detection	2013	[34]
R-G-NIR	ADC-lite camera Tetracam	Octocopter	Mikrokopter Okto	0.5	Nerello Mascalese	variability monitoring	2013	[56]
R-G-B-NIR	ADC-lite camera Tetracam	Hexacopter	Mikrokopter Hexa-II	1.2	Cabernet Sauvignon	variability monitoring	2013	[58]
TIR	Miricle 307 K	Fixed wings	Viewer	11	Pinot noir	variability monitoring	2014	[71]
R-G-B-NIR	Canon PowerShot A480 (Canon U.S.A, New York, NY, USA)	Hexacopter	Hawkeye	1.9	Tempranillo	variability monitoring	2014	[59]
NIR	NA	Fixed wings	Sensefly eBee	14	NA	crop features detection	2015	[45]
R-G-B-NIR	MCA 6 Tetracam	Quadcopter	RPAS Md4-1000	5	Tempranillo	variability monitoring	2015	[60]
R-G-B-NIR	GP Hero 3 and Micro-Hyperspec A-Series (Headwall Photonics, MA, USA)	Octocopter	OnyxStar BAT-F8	NA	Nemea-Agiorgitiko	rows geometry estimations	2015	[78]
R-G-B-NIR	ADC-lite camera Tetracam	Hexacopter	Mikrokopter Hexa-II	NA	NA	crop features detection	2015	[44]
R-G-B	Pentax A40	NA	NA	2.5	Cencibel-Airén	rows geometry estimations	2015	[80]
R-G-B	Canon IXUS 220 HS	Fixed wings	senseFly Swinglet CAM	12	NA	rows geometry estimations	2015	[87]
TIR	EasIR-9	Quadcopter	HKPilotMega 2.7	NA	Carménère	variability monitoring	2016	[72]
R-G-B-NIR	ADC-Snap Tetracam	Octocopter	Mikrokopter Okto	2.4	NA	variability monitoring	2016	[82]
R-G-B	NA	Multirotor-Fixed wings	NA	4	Languedoc	rows geometry estimations	2016	[83]
R-G-B	NA	Quadcopter	DJI Phantom 2	NA	Cabernet Sauvignon	crop features detection	2016	[49]
R-G-NIR	ADC-lite camera Tetracam	Octocopter	Mikrokopter Okto	1.2	Cabernet Sauvignon	disease detection	2016	[92]
TIR	FLIR TAU II 320	Octocopter	Mikrokopter Okto	7.5	NA	variability monitoring	2016	[73]
R-G-B-NIR	ADC-Snap Tetracam	Octocopter	Mikrokopter Okto	8.5	tempranillo	variability monitoring	2017	[61]
R-G-B	Lumix DMC-FT4	Quadcopter	NA	NA	Carménère	crop features detection	2017	[35]
R-G-B-NIR	Coolpix P7700-ADC-lite camera Tetracam	Octocopter	DJI s1000	0.5	Sagiovese	variability monitoring	2017	[62]
R-G-B-NIR-RE	RedEDGE Micasense	Fixed wings	long range DT-18	3.1	Sauvignon–Colombard -Gamay-Duras	disease detection	2017	[93]
R-G-B	Coolpix P7700 camera	Octocopter	DJI s1000	NA	Sangiovese	crop features detection	2017	[48]
R-G-B	DMC-GF3	Fixed wings	NA	23.2	Nebbiolo	rows geometry estimations	2017	[88]
R-G-B-NIR-RE	MCA-6 Tetracam	Octocopter	Mikrokopter Okto	NA	Carmeneré	variability monitoring	2017	[74]
R-G-NIR	ADC-lite camera Tetracam	Octocopter	Mikrokopter Okto	0.4	Sangiovese-Petit Verdot– Cabernet Sauvignon	variability monitoring	2017	[63]
R-G-B	Olympus PEN E-PM1	Quadcopter	MD4-1000	1.1	Merlot-Albariño-Chardonnay	rows geometry estimations	2018	[84]
R-G-B-NIR-RE	Parrot Sequoia	NA	NA	2.5	NA	rows geometry estimations	2018	[90]
R-G-B-NIR-TIR	Canon EOSM10-tetracam ADC Snap -FLIR TAU II 320	Hexacopter	Mikrokopter	10.3	Sangiovese	variability monitoring	2018	[64]
R-G-B	DJI FC6310	Quadcopter	DJI Phantom 4	NA	NA	crop features detection	2018	[46]
R-G-B-NIR- Hyp.	Canon 5DsR-R0Edge MicaSense Nano-Hyperspec	Hexacopter	S800 EVO Hexacopter	11.7	Chardonnay-Pinot Noir Shiraz-Merlot-Cabernet Sauvignon-Roussanne	disease detection	2018	[95]
R-G-B	DJI FC6310	Quadcopter	DJI Phantom 4	0.9	NA	rows geometry estimations	2018	[79]
R-G-B-NIR-TIR	Micro MCA-6 Tetracam-FLIR TAU2	Octocopter	Mikrokopter Okto	NA	Cabernet Sauvignon	variability monitoring	2018	[75]
R-G-B-NIR-RE	Parrot Sequoia	NA	NA	1.5	Nebbiolo	rows geometry estimations	2019	[89]
R-G-B	SONY α ILCE-5100L	Quadcopter	microUAV md4-1000	5	Syrah	disease detection	2019	[94]
R-G-B-NIR-RE	Parrot Sequoia	NA	NA	2.5	Nebbiolo	prescription mapping	2019	[98]
R-G-B-NIR	ADC-Snap Tetracam	Octocopter	Mikrokopter Okto	7.5	tempranillo	variability monitoring	2019	[66]
R-G-B-NIR	Olympus PEN E-PM1-SONY ILCE-6000	Quadcopter	MD4-1000	1	Pedro Xime’nez	crop features detection	2019	[39]
R-G-B-TIR	DJI FC6310-Optris PI450	Quadcopter	DJI Phantom 4 pro	1.8	Sangiovese –Petit Verdot – Cabernet Sauvignon	variability monitoring	2019	[76]
R-G-B-NIR-RE	RedEDGE Micasense	Hexacopter	UAVHEXA	5	Merlot	prescription mapping	2019	[97]
R-G-B-NIR-RE-TIR	Parrot Sequoia-thermoMAP	Quadcopter-Fixed wings	DJI Phantom 4-Sensefly eBee	0.3	Malvasia Fina	variability monitoring	2019	[65]
R-G-B	NA	multirotor	NA	11.3	Syrah-Grenache	crop features detection	2019	[50]
R-G-B-NIR	ADC-Snap Tetracam-ThermalCapture FUSION	Hexacopter	Mikrokopter	2.4	Barbera-Sangiovese	crop features detection	2019	[47]
R-G-B	Olympus PEN E-PM1	Quadcopter	MD4-1000	0.9	Merlot and Albariño	scrop features detection	2020	[41]
R-G-B	NA	Quadcopter	Parrot Bebop 2	1	NA	rows geometry estimations estimations	2020	[85]
R-G-B-NIR-RE	RedEDGE Micasense	Hexacopter	UAVHEXA	17.7	Chardonnay-Merlot-Cabernet Sauvignon	prescription mapping	2020	[99]
R-G-NIR-RE	Mapir survey 3	Quadcopter	DJI Phantom 4 pro	1.3	Cagnulari	rows geometry estimations estimations	2020	[86]
R-G-B-NIR-RE-TIR	Parrot Sequoia–thermoMAP senseFly	Quadcopter-Fixed wings	DJI Phantom 4	2.1	Alvarinho-Loureiro	variability monitoring	2020	[67]
R-G-B-NIR-RE	Parrot Sequoia	NA	NA	2.5	Nebbiolo	rows geometry estimations	2020	[91]
R-G-B-NIR-RE	Parrot Sequoia	NA	NA	2.5	Nebbiolo	rows geometry estimations	2020	[81]

Legend: NA—Not available; R—Red; G—Green; B—Blue; NIR—Near infrared; TIR—Thermal infrared; Hyp—Hyperspectral.

## Data Availability

Not applicable.

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
