# Peer review of "Advances in Unmanned Aerial System Remote Sensing for Precision Viticulture"

_sensors, 2021, doi:10.3390/s21030956_

Round 1

Reviewer 1 Report

The authors present a review of UAV-based remote sensing in precision viticulture. The review focuses on the analysis of remote sensors, data extraction and analysis methods. This is an important area of research for improving management and output of vineyards, which make up a considerable portion of the agricultural sector. Therefore, it is my opinion that this work is important, interested and appropriate for Sensors with minor revision.

Review Comments and Suggestions

In line 72, the authors state the following, “The use of drones in agricultural scenarios is now well established, but currently there is no state of the art regarding their use in venti culture”. Additional information is needed to support the claim made in the first clause of the sentence. For example, providing a percentage of agricultural operations incorporating UAVs with appropriate references would be helpful. The second clause of the sentence also suggests that there are no new developments in UAV-based technology for viticulture.  However, a review article published in 2015 by Matese et al., and references therein, highlight viticulture UAV applications. Therefore, the authors need to reconcile their statement with existing published work.

Matese, Alessandro, and Salvatore Filippo Di Gennaro. "Technology in precision viticulture: A state of the art review." International journal of wine research 7 (2015): 69-81.

In line 100, the authors reference 2G_RBi spectral index results from [26]. Please consider providing additional details about what the 2G_RBI spectral index quantifies specifically.

In line 468, the authors draw observations about the use of fixed-wing and multirotor UAVs in viticulture applications. However, besides the information presented in Table 1, the text does not present a review of fixed-wing and multirotor flight characteristics.  Please include a paragraph discussing how fixed-wing and multirotor UAVs flight characteristics are useful for precision viticulture applications.

In parallel to the exploration of UAV-based technology for viticulture monitoring, research efforts have also investigated the use of UAVs for atmospheric sensing. Considering that multiple studies have highlighted the importance of micrometeorological parameter observations for microclimate modeling in vineyards, there is an opportunity to bridge the gap between atmospheric sensing and viticulture applications of UAVs in the discussion section of this manuscript (see references below).

Orlandini, S., G. Zipoli, B. Gozzini, E. Egger, E. Marinelli, and P. Storchi. "Micrometeorology of vineyards and phytopathological models 1." EPPO Bulletin 21, no. 3 (1991): 431-439.

Andreoli, Valentina, Davide Bertoni, Claudio Cassardo, Silvia Ferrarese, Caterina Francone, and Federico Spanna. "Analysis of micrometeorological conditions in Piedmontese vineyards." (2018): 27-40.

Matese, Alessandro, Alfonso Crisci, Filippo Salvatore Di Gennaro, Edoardo Fiorillo, Jacopo Primicerio, Piero Toscano, Francesco Primo Vaccari, Stefano Di Blasi, and Lorenzo Genesio. "Influence of canopy management practices on vineyard microclimate: definition of new microclimatic indices." American journal of enology and viticulture 63, no. 3 (2012): 424-430.

Barbieri, Lindsay, Stephan T. Kral, Sean CC Bailey, Amy E. Frazier, Jamey D. Jacob, Joachim Reuder, David Brus et al. "Intercomparison of small unmanned aircraft system (sUAS) measurements for atmospheric science during the LAPSE-RATE campaign." Sensors 19, no. 9 (2019): 2179.

Author Response

Reviewer 1

Comments and Suggestions for Authors

The authors present a review of UAV-based remote sensing in precision viticulture. The review focuses on the analysis of remote sensors, data extraction and analysis methods. This is an important area of research for improving management and output of vineyards, which make up a considerable portion of the agricultural sector. Therefore, it is my opinion that this work is important, interested and appropriate for Sensors with minor revision.

Review Comments and Suggestions

In line 72, the authors state the following, “The use of drones in agricultural scenarios is now well established, but currently there is no state of the art regarding their use in venti culture”. Additional information is needed to support the claim made in the first clause of the sentence. For example, providing a percentage of agricultural operations incorporating UAVs with appropriate references would be helpful. The second clause of the sentence also suggests that there are no new developments in UAV-based technology for viticulture.  However, a review article published in 2015 by Matese et al., and references therein, highlight viticulture UAV applications. Therefore, the authors need to reconcile their statement with existing published work.

Matese, Alessandro, and Salvatore Filippo Di Gennaro. "Technology in precision viticulture: A state of the art review." International journal of wine research 7 (2015): 69-81.

We provided some information about the most common agricultural operations which incorporate UAVs with the related reference (lines 71-77). We also include the forecast of SESAR about the development and growth of UAV agricultural market in Europe. It would help the readers to understand the future improvement and development of drone sector in the agricultural field. The aim of the review is not to give a state of the art of technology in precision viticulture but to focus on the analysis of the used sensors, the data extraction, analysis methods, and discusses the potential of UAV’s remote sensing as a management tool in viticulture scenarios. We did not find any review which focus solely on this specific theme. The mentioned review gives useful and well argumentized information about the technology applied in precision viticulture, including a general paragraph on UAV in viticulture. Anyway, e modified the part in order to clarify this point and warn the reader about the aim of the review (lines 77-80).

In line 100, the authors reference 2G_RBi spectral index results from [26]. Please consider providing additional details about what the 2G_RBI spectral index quantifies specifically.

In lines 104-106 we have reported some additional information about the characteristics of the index. We believe that the review aims to illustrate in a general way the advantages inherent in the use of the index without going too much into the specifics of its nature. If the reader is interested in deepening his knowledge about the application of the difference index for the segmentation of rows (as described), he can refer to the paper mentioned.

In line 468, the authors draw observations about the use of fixed-wing and multirotor UAVs in viticulture applications. However, besides the information presented in Table 1, the text does not present a review of fixed-wing and multirotor flight characteristics.  Please include a paragraph discussing how fixed-wing and multirotor UAVs flight characteristics are useful for precision viticulture applications.

We included some general information about fixed wing and multirotor, about advantage and disadvantage of their use. We also discussed how these systems can be useful fot precision viticulture purposes. In order to not affect the fluidity of the manuscript we chose to don’t write a new paragraph but to include this information at the beginning of the Results and Discussion part (lines 473-491).

In parallel to the exploration of UAV-based technology for viticulture monitoring, research efforts have also investigated the use of UAVs for atmospheric sensing. Considering that multiple studies have highlighted the importance of micrometeorological parameter observations for microclimate modeling in vineyards, there is an opportunity to bridge the gap between atmospheric sensing and viticulture applications of UAVs in the discussion section of this manuscript (see references below).

Orlandini, S., G. Zipoli, B. Gozzini, E. Egger, E. Marinelli, and P. Storchi. "Micrometeorology of vineyards and phytopathological models 1." EPPO Bulletin 21, no. 3 (1991): 431-439.

Andreoli, Valentina, Davide Bertoni, Claudio Cassardo, Silvia Ferrarese, Caterina Francone, and Federico Spanna. "Analysis of micrometeorological conditions in Piedmontese vineyards." (2018): 27-40.

Matese, Alessandro, Alfonso Crisci, Filippo Salvatore Di Gennaro, Edoardo Fiorillo, Jacopo Primicerio, Piero Toscano, Francesco Primo Vaccari, Stefano Di Blasi, and Lorenzo Genesio. "Influence of canopy management practices on vineyard microclimate: definition of new microclimatic indices." American journal of enology and viticulture 63, no. 3 (2012): 424-430.

Barbieri, Lindsay, Stephan T. Kral, Sean CC Bailey, Amy E. Frazier, Jamey D. Jacob, Joachim Reuder, David Brus et al. "Intercomparison of small unmanned aircraft system (sUAS) measurements for atmospheric science during the LAPSE-RATE campaign." Sensors 19, no. 9 (2019): 2179.

We appreciate the suggestion given by the reviewer about how UAV application can bridge micrometeorological parameter observations for microclimate modelling in vineyard. It could be an interesting topic for further studies which would improve crop protection, highlight the damage to the ecosystem and help farmers to reduce the numbers of treatments (as stated in the manuscript).The new part has been added at the end of the Results and Discussion part as suggested (lines 531-556).

Reviewer 2 Report

The paper presents a review on the use of UAVs for precision viticulture. The structure of the manuscript is correctly designed and bibliographic references are adequately and homogeneously distributed along the text. Table 1, providing an overview of the different UAVs and sensors used for PV and other information derived from the literature. This table represents a good support for the readers and summarizes the main information reported in the manuscript. I would suggest an extension of section 4 (Conclusions and remarks) especially in the definition of further perspective in this field of research; this topic is only partially addressed at the current state of the work. Also general remarks could be better addressed and described.
English language is fine and the work is easy to read and understand (line 84. check the use of "guaranty", verbal form guarantee could be more appropriate in this sentence)
A final consideration on the use of both the terms UAV and Drone. I generally prefer the use of only one definition, and UAV is, in my opinion, more appropriate than drone (that generally reminds a military context).

Author Response

Reviewer 2

Comments and Suggestions for Authors

The paper presents a review on the use of UAVs for precision viticulture. The structure of the manuscript is correctly designed and bibliographic references are adequately and homogeneously distributed along the text. Table 1, providing an overview of the different UAVs and sensors used for PV and other information derived from the literature. This table represents a good support for the readers and summarizes the main information reported in the manuscript. I would suggest an extension of section 4 (Conclusions and remarks) especially in the definition of further perspective in this field of research; this topic is only partially addressed at the current state of the work. Also general remarks could be better addressed and described.

We have extended Section 4 by including more insights into future prospects, and also some general considerations.

English language is fine and the work is easy to read and understand (line 84. check the use of "guaranty", verbal form guarantee could be more appropriate in this sentence)

As suggested, we changed the verbal form from "to guaranty" to "to guarantee" (line 88).

A final consideration on the use of both the terms UAV and Drone. I generally prefer the use of only one definition, and UAV is, in my opinion, more appropriate than drone (that generally reminds a military context).

We have replaced the term "drone" with "UAV" and "drones" with "UAVs". The term drone hints at a military use and using the term UAV throughout the manuscript is more useful and facilitates understanding.

Reviewer 3 Report

Sassu et al. provided a thorough review of using unmanned aerial system data for precision viticulture. In view of the great potential for UAS research on agricultural studies, this study is important and relevant. This paper is in general well written and well designed. It summarized the major breakthrough of UAS studies in recent years. However, there are several major issues in this manuscript. This manuscript requires a major revision to improve quality. Here are some comments that may be helpful to improve the manuscript.

  1. Unmanned Aerial Vehicle is not a standard word for Unmanned Aerial Vehicle based remote sensing. It is better to use "unmanned aerial system", "remotely piloted aerial system", or "unoccupied aerial system".
  2. In the title, the keyword on review is missing. It is better to tell readers that this paper is a review article more straightforward. I suggest that the title could be "Advances in Unmanned aerial system remote sensing for precision viticulture".
  3. The differences between subtitles are not clear. "Vineyard variability monitoring" is similar to "crop feature detection"
  4. The paper has done a good summary of current research work. However, as a review paper, it is also important to point out what is missing in past research and what is needed for future research. More recent and relevant studies on using UAS remote sensing for soil moisture monitoring, water consumption (evapotranspiration), surface energy budget, water use efficiency, and ecosystem gross productivity should be added to enrich the future research directions.

Author Response

Reviewer 3

Comments and Suggestions for Authors

Sassu et al. provided a thorough review of using unmanned aerial system data for precision viticulture. In view of the great potential for UAS research on agricultural studies, this study is important and relevant. This paper is in general well written and well designed. It summarized the major breakthrough of UAS studies in recent years. However, there are several major issues in this manuscript. This manuscript requires a major revision to improve quality. Here are some comments that may be helpful to improve the manuscript.

  1. Unmanned Aerial Vehicle is not a standard word for Unmanned Aerial Vehicle based remote sensing. It is better to use "unmanned aerial system", "remotely piloted aerial system", or "unoccupied aerial system".

We prefer to use the term Unmanned Aerial System (UAV) because it is the most widely used and known. In the papers we examined, most of the times this apellative is used, so we think it can be the best solution also for those who make a bibliographical research on the theme.

  1. In the title, the keyword on review is missing. It is better to tell readers that this paper is a review article more straightforward. I suggest that the title could be "Advances in Unmanned aerial system remote sensing for precision viticulture".

We opted to replace the title of the review (line 2) with the one suggested by the reviewer "Advances in Unmanned Aerial Vehicle Remote Sensing for Precision Viticulture". We think his suggestion may help the readers to perceive the nature of the review and give it a more appealing title.

  1. The differences between subtitles are not clear. "Vineyard variability monitoring" is similar to "crop feature detection"

We chose to change the subtitle 2.1 “Crop features detection” (line 85) in “Rows segmentation and crop features detection techniques”. The aim of this section is to describe the multiples techniques used from the research teams to segment the vine rows from the soil and the herbaceous crops in the inter-rows and to detect the features useful for further studies. We furthermore opted to change the subtitle 2.2 “Vineyard variability monitoring” (line 173) to “Vineyard remote analysis for variability monitoring”. The aim of this paragraph is to describe the methodology used to analyse the variability of physical, chemical, biological variables related to the productivity of vineyards.

  1. The paper has done a good summary of current research work. However, as a review paper, it is also important to point out what is missing in past research and what is needed for future research. More recent and relevant studies on using UAS remote sensing for soil moisture monitoring, water consumption (evapotranspiration), surface energy budget, water use efficiency, and ecosystem gross productivity should be added to enrich the future research directions.

The new part has been added close to the end of the Results and Discussion part (lines 525-530), as an addition to the absence of works on another topic (spraying application). Vineyard UAV remote sensing could be an interesting approach to improve soil feature knowledge and vineyard site-specific management (as stated in the manuscript).

Round 2

Reviewer 3 Report

The authors have addressed my comments and suggestions. The current format has significantly improved. However, there are still a few problems and this manuscript requires further revision.

  1. UAV only refers to the vehicle and is not a scientific term to use in this manuscript. The terminology "Unmanned aerial system, UAS" is more appropriate and suitable.
  2. How about the UAV data quality progress for ecosystem monitoring? How do you deal with UAV with cloud shadow issues?
  3. UAVs can only provide snapshots monitoring of land surface conditions. How do you monitor ecosystem conditions when UAV imagery is absent?
  4. The font size of Table 1 is too small.

Author Response

Reviewer 3

Comments and Suggestions for Authors

The authors have addressed my comments and suggestions. The current format has significantly improved. However, there are still a few problems and this manuscript requires further revision.

  1. UAV only refers to the vehicle and is not a scientific term to use in this manuscript. The terminology "Unmanned aerial system, UAS" is more appropriate and suitable.

We replaced the term "UAV" with "UAS" and "UAVs" with "UASs". We also changed the title in “Advances in Unmanned Aerial System Remote Sensing for Precision Viticulture”.  Since, as stated in the previous review, the term UAV is more widely known and used in the works cited by the manuscript, we reserve the right to leave it among the key words. Its presence will facilitate user research until the term UAS will be better established.

  1. How about the UAV data quality progress for ecosystem monitoring? How do you deal with UAV with cloud shadow issues?

This particular aspect of UAS remote sensing is a common issue, so we insert some additional information in the introduction paragraph. (lines 57-63).

  1. UAVs can only provide snapshots monitoring of land surface conditions. How do you monitor ecosystem conditions when UAV imagery is absent?

In lines 564-573, we have discussed abundantly the use of UAS as integrative systems within a decision support system that requires hourly extracted data. In each case we have added a section stating the need for manual ground sampling and extrapolation of data from proximal sensing between flights. We hope we understood the point of the suggestion.

  1. The font size of Table 1 is too small

We enlarged the font size to 7. A bigger size would not allow the presence of all the columns and the relative information in an appropriate format. The deleting of one of the columns (as the Objective one) would implicate a loose of information we prefered to avoid.
